# Learning 3D Dense Correspondence via Canonical Point Autoencoder

**An-Chieh Cheng**[1]**, Xueting Li**[2]**, Min Sun**[13]**, Ming-Hsuan Yang**[245]**, Sifei Liu**[6]**,**
[1]National Tsing-Hua University  [2]University of California, Merced,
[3]Joint Research Center for AI Technology and All Vista Healthcare,
[4]Google Research,  [5]Yonsei University,  [6]NVIDIA

## Abstract

We propose a canonical point autoencoder (CPAE) that predicts dense correspondences between 3D shapes of the same category. The autoencoder performs two key functions: (a) encoding an arbitrarily ordered point cloud to a canonical primitive, e.g., a sphere, and (b) decoding the primitive back to the original input instance shape. As being placed in the bottleneck, this primitive plays a key role to map all the unordered point clouds on the canonical surface and to be reconstructed in an ordered fashion. Once trained, points from different shape instances that are mapped to the same locations on the primitive surface are determined to be a pair of correspondence. Our method does not require any form of annotation or self-supervised part segmentation network and can handle unaligned input point clouds within a certain rotation range. Experimental results on 3D semantic keypoint transfer and part segmentation transfer show that our model performs favorably against state-of-the-art correspondence learning methods. The source code and trained models can be found at https://anjiecheng.github.io/cpae/.

## 1 Introduction

With prior knowledge and experience, humans can easily perceive corresponding object parts (e.g., the wings from two different airplanes), understand their shape and appearance variance, in order to distinguish different objects coming from the same category. In computer vision, modeling dense correspondence between 3D shapes in one category is fundamental for numerous applications, such as robot grasping [1, 2], object manipulation [3] and texture mapping [2, 4]. However, existing 3D cameras typically capture raw point clouds of shape surfaces that are arbitrarily-ordered and unstructured, in which correspondences are not established. 3D mesh representation, although is usually parameterized with UV maps that can indicate correspondences, cannot be directly obtained from sensors and needs to be reconstructed from other types of representations, e.g., 2D images [5, 6] or 3D point clouds [7]. In this work, we focus on learning point cloud correspondences, which remains an open challenge since it is infeasible to label ground truth correspondence annotations.

Without ground truth annotations, existing methods mainly discover shape correspondences via seeking a form of canonical space that can associate various instance shapes. For example, in particular shape domains such as human bodies [8] and human faces [9], parameterized shape primitives have been designed to fit the observed raw data and to obtain the correspondences. Such designs, however, cannot be generalized to other categories, e.g., man-made objects [10]. Recently, several part co-segmentation networks relax the requirements of specific parameterized primitives, but instead decompose input shapes into an ordered group of simplest part constitutions [11, 12], in a self-supervised manner. These methods, however, require careful selection of the autoencoder architectures (i.e., they need to be considerably shallow to let the branches only able to represent simple shapes), and the number of part bases. Moreover, such part-based representation does not explicitly provide fine-grained (e.g., point-level) correspondences.

35th Conference on Neural Information Processing Systems (NeurIPS 2021).

In this work, we introduce a novel canonical space where dense (i.e., point-level) correspondences for all the shapes of a category can be explicitly obtained from. Inspired by 3D mesh representation [5, 13, 14, 15] where shapes from one category are represented as deformations on top of a shape primitive, in our work, we set the canonical space as a 3D UV sphere. Our goal is to learn a "point cloud-to-sphere mapping" such that corresponding parts from different instances overlap when mapped onto the canonical sphere. In other words, similar to the mesh representation, a unique UV coordinate can represent the same semantic point/local region of shapes (e.g., the tip of an aeroplane's wing), regardless of shape variations. Towards this goal, we introduce the canonical point autoencoder (CPAE): we place the sphere primitive at the bottleneck; the encoder non-linearly maps each individual input shape to the sphere primitive, where the decoder deforms the primitive back to match the original shape. We show that with several self-supervised objectives, this autoencoder architecture effectively (1) enforces the input points warped to the surface of the sphere primitives, and (2) encourages those corresponding points from different instances mapped to the same location on the sphere – both guarantee that the network learns correct dense correspondences. Essentially, we *do not* assume all object shapes in one category having the same topology, e.g., an armchair does not have correspondence on its armrests, with another instance without an armrest. To introduce such uncertainty for correspondence matching, we propose an adaptive Chamfer loss on the bottleneck to allow customized primitive for each instance. As such, we are able to determine if a point on one instance has a correspondence in another point cloud.

One advantage of the proposed method compared to the recent work [12] is that we can learn correspondences even when instances in the training dataset are not aligned, i.e., our model is rotation-invariant within a certain rotation range and does not need to predict an additional rotation matrix as used in [12]. The main contributions of this work are:

- We introduce a novel canonical space – a UV sphere, that explicitly represents dense correspondences of shapes from one category.
- With the canonical space on the bottleneck, we design an autoencoder that learns such a "point cloud-to-sphere mapping" via a group of self-supervised objectives.
- We apply the proposed method on various categories and quantitatively evaluate on the task of 3D semantic keypoint transfer and part segmentation label transfer, achieving comparable if not better performance than state-of-the-art methods.

## 2 Related Work

**Deep Learning on Point Clouds.**   As a flexible and memory efficient representation of 3D shapes, point cloud has been widely studied and combined with deep neural networks. The PointNet [16] solves point cloud classification and segmentation by using MLPs and a max pooling layer to aggregate 3D shape information. One crucial property of the PointNet is that it is able to handle unordered input and is thus invariant to point permutations. In our work, we utilize the first few layers of the PointNet as our shape encoder. Another line of works [17, 18, 19, 20] focus on reconstructing or generating point clouds. The most representative and related to our work is the FoldingNet [17], where a point cloud is first encoded by a graph-based encoder and then reconstructed by sequentially applying the "folding operation" (instantiated by MLPs) to a 2D UV map. In this work, we reveal why the MLPs are able to preserve point order and utilize it as building blocks in our CPAE.

**Primitive Fitting.**   Primitive fitting has a long history in computer vision [21]. Several approaches focus on fitting individual samples with shape priors, such as blocks world [21], generalized cylinders [22], geons [23], Lego pieces [24], qualitative 3D blocks [25], and patches [26]. However, these primitives are analyzed per-instance without order, therefore, cannot be used for the correspondence. Some other approaches focus on fitting over a collection of shapes, which provide ordered set of primitives [11, 27, 28, 29, 30, 31, 32]. However, most of these approaches can only infer part-level correspondences and require additional supervision such as the number of primitives. In this work, we directly learns set-to-set alignment between point clouds, thus requires no shape prior.

**3D Dense Correspondence.**   Given a pair of source and target instance in the same category, 3D dense correspondence learning targets at finding a corresponding point in the target instance for each point from the source instance. Several approaches [33, 34, 35] resolve this task through point cloud registration, using labeled pairwise correspondence as supervision. To relax constraints on supervision, Bhatnagar et al. [36] predict part correspondences to a template via implicit functions. Unfortunately, they require part labels for training. To unsupervisedly learn 3D dense correspondence,

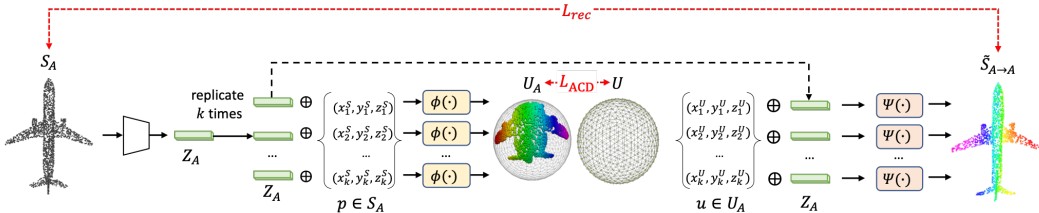

Figure 1: Overview of the proposed method. Here, $p = (x_i^S, y_i^S, z_i^S)$ is a point on the input point cloud $S_A$, $u = (x_i^U, y_i^U, z_i^U)$ is a point on the primitive $U_A$. The "$\oplus$" sign indicates concatenation. $\Phi(\cdot)$ and $\Psi(\cdot)$ are MLPs as discussed in Section 3.1.

Chen et al. [37] propose a method to learn 3D structure points that are consistent across different instances. However, the model assumes structure similarity among different instances, which ignores intra-class variants and fails to detect non-existing correspondences between dissimilar shapes in the same category. Most relevant to our work, Liu et al. [12] introduce an unsupervised approach that leverages part features learned by the BAE-NET [11] to build dense correspondences. Their algorithm is able to calculate a confidence score representing the probability of correspondence. However, in order to train the implicit function, they require additional knowledge of object surface to compute the occupancy. Moreover, the correspondence learning in [12] heavily relies on the completeness of the part features from the BAE-NET [11], which can lead to incorrect correspondences for parts that BAE-NET cannot separate (e.g., flat surfaces, objects with fine-grained details). In contrast, our approach directly learns dense correspondences from the point cloud with self-supervision.

**Cycle Consistency.**  Our cross reconstruction loss is related to literature that uses cycle consistency as a supervisory signal to learn correspondence without ground truth annotations. Some of these works exploit the consistency among different modalities (e.g., 2D-3D). Specifically, [38] aim to learn cross-instance pairwise correspondence for 2D images using cycle-consistency terms guided by 3D synthetic data. [39] and [40] both leverage cycle-consistency terms to enforce correspondence between a 2D image and a 3D canonical template. Aside from these works that focus on cross-modality, work from [41] is mostly related to ours. They propose a "point-wise" cycle consistency loss that explicitly enforces a point deformed through any cycle of deformations to be mapped back to the origin location. Different from their approach, our consistency is instance-wise and we do not assume bijection mapping among shapes.

**3D Deformable Mesh Representations.**  Our algorithm is also related to methods [5, 13, 14, 15] that represent instance shapes as mesh deformations of a mesh primitive, i.e. a sphere template. Vertices on each instance surface that are mapped to the same locations on the shape primitive are discovered as correspondences. One obvious limitation is that only genus 0 shapes, e.g., birds [5, 13] can be deformed from a mesh sphere. In contrast, ours does not have such restriction thanks to the proposed non-linear mappings, instead of explicit deformation.

## 3   Proposed Method

In this section, we introduce our end-to-end CPAE for learning dense correspondences from point clouds without ground truth annotation. Given a point cloud $S_A$ with individual point $p \in \mathbb{R}^3$ (see Figure 1), our model: (1) predicts its canonical primitive $U_A$ (a "deformed" point cloud with the same number of points as $S_A$), which is supposed to be as close as possible to the canonical sphere in the bottleneck; (2) reconstructs the original input point cloud back from $U_A$. We show that while the first ensures each input point cloud to be warped to the surface of the sphere primitive, the second indirectly encourages the corresponding points from different point cloud instances to overlap during mapping to the primitive. In the following, we first describe our network architecture, i.e., the encoder and decoder modules. We then introduce the adaptive Chamfer loss and the reconstruction losses that are applied to each individual module. Finally, we show that our decoder reconstructs ordered point clouds, i.e., different point cloud instances can fetch their correspondences directly via the point indices, which provides a more accurate inference.

## 3.1 Network Architecture

Our method learns two mapping functions: one maps each individual 3D point $p \in \mathbb{R}^3$ in the world coordinates to the canonical space $\Phi(p) = u, u \in \mathbb{R}^3$, while the other conducts the inverse mapping $\Phi^{-1}(u) = p$. Instead of using a reversible network, we instantiate $\Phi(\cdot)$ and $\Phi^{-1}(\cdot)$ with two MLPs respectively, and cascade them to construct an autoencoder (i.e., $\Phi(\cdot)$ is the encoder and $\Psi(\cdot) \approx \Phi^{-1}(\cdot)$ is the decoder). However, this mapping function cannot be generalized to different point clouds if it only receives the coordinate of a single point $p$ as the input. Therefore, we formulate them as conditional mapping functions by concatenating the input with a shape latent code $Z_A$ that represents a unique input shape $S_A$, e.g., $\Phi(p, Z_A)$, in order to generalize the mapping functions to all instances in the same category.

There are three key modules in the proposed method: (1) a PointNet encoder that produces the shape latent code; (2) an encoder MLP (denoted as *canonical* mapping) that maps a point cloud to a primitive sphere, and (3) a decoder MLP (denoted as *inverse* mapping) that deforms the primitive sphere back to the input point cloud.

**PointNet Encoder.** As shown in Figure 1, given a point cloud $S_A \in \mathcal{R}^{k \times 3}$ with $k$ points, we encode it using a PointNet [16] encoder in a way similar to [8, 18, 19]. Each 3D point of the input point cloud is represented as a 512 dimensional vector using an MLP with 3 hidden layers of 64, 128, 512 neurons and ReLU activations. We then aggregate all point features with max-pooling and a linear layer to generate a global latent code $z_A \in \mathcal{R}^{512}$. We use PointNet because it produces a robust latent code representing the global shape, which is also invariant to input permutation.

**Canonical Mapping $\Phi(\cdot)$.** To learn the canonical mapping function, we concatenate each point with the global latent code $[p, Z] \in \mathbb{R}^{515}$ as input to the MLP, which then outputs a 3D coordinate $u \in \mathbb{R}^3$ in the canonical space. We construct a sphere point cloud by uniformly sampling a large number of points from a standard sphere mesh, and place them as the canonical primitive at the bottleneck. A Chamfer loss is used to measure the difference between the outputted point $u$ and primitive to encourage the mapped point adhering to the surface (see Eq. (1)).

Ideally, all the mapped instances from the same category should be as close as possible to the canonical sphere primitive. However, an instance may include parts/regions that do not exist in other instances due to intra-class variation, e.g., not all chairs include armrests. With a conventional Chamfer loss, the shapes for objects with rare components do not converge during training, i.e., those rare components are mapped to locations that are far away from the primitive surface. Thus, we relax the bidirectional constraint of the Chamfer loss and allow each instance to produce its own "instance primitive" to some extent (see $U_A$ in Figure 1 ). This is formulated via an adaptive Chamfer loss $L_{ACD}$:

$$L_{ACD}(U_A, U) = \frac{1}{|U_A|} \sum_{p \in U_A} \min_{q \in U} \|p - q\|_2 + \alpha \frac{1}{|U|} \sum_{q \in U} \min_{p \in U_A} \|q - p\|_2 \tag{1}$$

where $\alpha \sim [0, 1]$ is an adaptive parameter indicating to what extent the predicted instance primitive should match a canonical sphere, $U_A$ and $U$ are the instance primitive and the canonical UV primitive (e.g., a 3D UV sphere), respectively. When $\alpha = 1$, $L_{ACD}$ is equivalent to calculating the conventional Chamfer distance between the unfolded primitive and the canonical UV primitive. During training, $\alpha$ is initialized to 1 and gradually decreased to 0. This allows the canonical mapping to predict instance-aware primitive since the second term of the Chamfer loss no longer constraints the primitive to be consistent with a canonical sphere. In the experiments (see Section 4.5), we show that this design allows us to infer rare object components that have no correspondences in other instances since non-corresponding regions usually occupy a distinct area in the canonical space.

**Inverse Mapping $\Psi(\cdot)$.** Similarly, we utilize another MLP in the decoder, which receives the concatenation of one point from the bottleneck and the global shape code, i.e., $[u, Z] \in \mathbb{R}^{515}$, as the input. The function learns to map it back to its original world coordinate $p$, which can be fulfilled via a point-to-point reconstruction loss. We leverage an MSE loss $L_{MSE}$, a Chamfer distance $L_{CD}$ and an Earth Mover Distance (EMD) $L_{EMD}$ between the reconstruction $\hat{S}$ and input point cloud $P$:

$$L_{rec}(P, \hat{S}) = \mu_1 L_{MSE}(P, \hat{S}) + \mu_2 L_{CD}(P, \hat{S}) + \mu_3 L_{EMD}(P, \hat{S}) \tag{2}$$

where $\mu_1, \mu_2, \mu_3$ are the weights, and empirically determined, $\mu_1 = 1e3$, $\mu_2 = 10$, and $\mu_3 = 1$.

The inverse mapping bears some resemblance to the principle of FoldingNet [17], which adopts an MLP architecture to map one 2D coordinate together with a global shape representation to the 3D

world[1]. However, since FoldingNet does not have the forward mapping function as ours and the input points are untraceable, it needs to use Chamfer loss, instead of a point-wise loss (Eq. (2)) as ours.

**Align Canonical Mappings via Cross-Reconstruction.** With the encoder and objective function, the canonical mapping is able to map the input points to the surface of the canonical primitive. However, even with the reconstruction loss (Eq. (2)), there is no guarantee that the corresponding points from different shape instances can be overlapped on the sphere – which is the key to learn dense correspondences. For instance, in Figure 1, while $S_A$ is mapped to the frontal half of the sphere, it is likely that another shape $S_B$ will be mapped to the rear half.

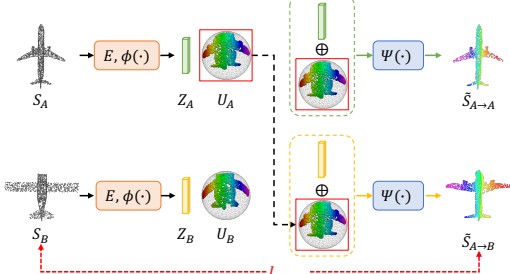

Figure 2: Cross-reconstruction. The $E, \Phi(\cdot), \Psi(\cdot)$ are identical as in Figure 1.

To better align the mapped points, we introduce a cross-structured decoder that leverages: (1) a self-reconstruction branch as presented in Eq. (2), and (2) a cross-reconstruction branch. As shown in Figure 2, we feed the combination of the predicted instance primitive $U_A$ of point cloud $S_A$, with the shape latent vector $Z_B$ of another randomly sampled point cloud $S_B$ to the folding decoder. We then minimize the Chamfer distance between the output shape $\hat{S}_{A \to B}$ and $S_B$. The cross-reconstruction loss between the point cloud $S_A$ and $S_B$ is:

$$L_{cross}(S_A, S_B) = L_{CD}(\hat{S}_{A \to B}, S_B) \tag{3}$$

The cross-reconstruction branch ensures even by swapping the predicted primitives $U_A$ and $U_B$, the inverse mapping can still reconstruct their own shapes, conditioned on their own shape latent codes. That is, the decoder encourages $U_A$ and $U_B$ to overlap on the canonical primitive, and thus ensures the network learns correct correspondences.

**Relation to Implicit Function.** We note that since both mapping functions $\Phi(\cdot)$ and $\Psi(\cdot)$ process each input point independently, they can be interpreted as conditional implicit functions, which are widely applied for 3D shape reconstruction [42, 43], view synthesis [44, 45], and recently for self-supervised 3D correspondence [12]. Such an interpretation reveals that the MLPs indeed learn continuous mappings that are feasible for interpolations, e.g., once the decoder MLP is learned, it is able to map any point on a continues sphere surface that is not necessarily among the mapped input $u$, to the world coordinate. E.g, one can sample points densely from the surface of the sphere to reconstruct a continuous surface or a 3D mesh.

### 3.2 Finding Correspondences from Ordered Point Clouds

In this section, we show that the CPAE is able to generate ordered point clouds (see the output point clouds in Figure 1). Compared to the methods obtaining correspondences by directly tracing overlapped points from the canonical space, our model that makes use of ordered output point cloud generates more accurate results during the inference stage.

**Reconstruction of Ordered Point Clouds.** We show that an MLP [17] decoder preserves the order of points in the canonical primitives (shown as colored points within the sphere), when mapping them back to individual shapes. Given two adjacent points $U_A$ and $U_B$ from the sphere, we denote $p_A = \Psi(U_A, Z)$ and $p_B = \Psi(U_B, Z)$ as their mapped points in the output space, where $Z$ is the latent shape code. Since $\Psi(\cdot)$ contains only linear projections and ReLUs (i.e., piece-wise linear), it is easy to be proved as a continuous function. As such, $p_A$ and $p_B$ will also be close to each other, e.g., in Figure 1, the output shape exhibits a smooth transition of color map similar to the colored canonical shape on the sphere. Thus, the order is maintained as opposed to being shuffled in the output space.

**Inference via CPAE.** Given a source shape $S_A$ and a query point $p_A \in S_A$, we target at searching the correspondence of the query point in a target shape $S_B$. We first compute the shape latent vectors for the two shapes as $z_A$ and $z_B$. The query point is then mapped to a point $u_A$ in the canonical space by the canonical mapping encoder. By feeding the concatenation of $u_A$ and $z_B$ to the inverse

---

[1]Unlike our method, it utilizes and samples points from a standard 2D UV map instead of a 3D sphere.

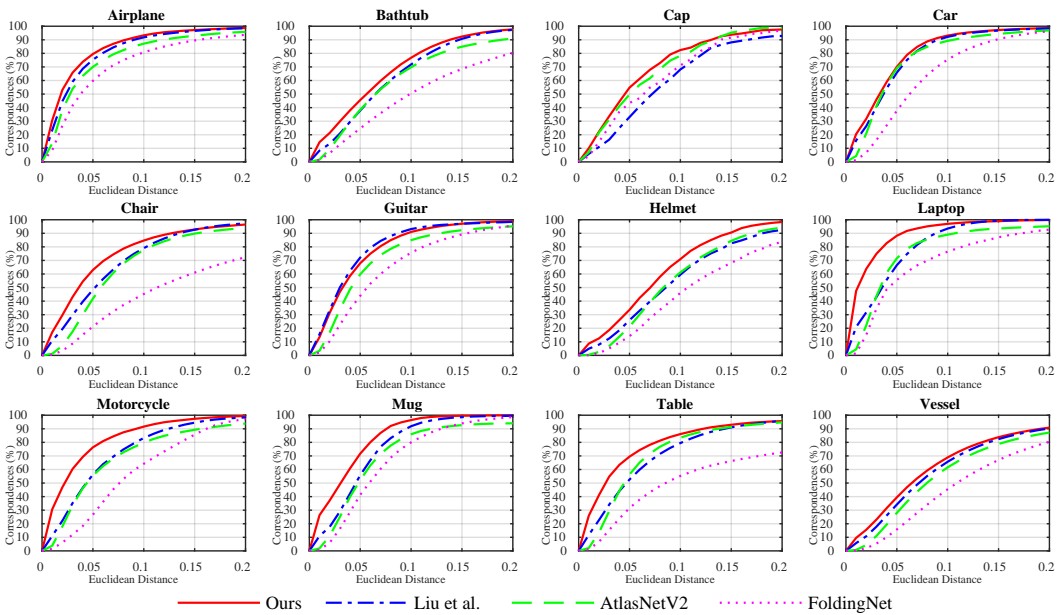

Figure 3: Correspondence accuracy for 12 categories in the KeypointNet dataset.

mapping decoder, we further map $u_A$ to a point $p_{A \to B}$ on the reconstructed target shape $S_{A \to B}$. The correspondence of $p_A$ (denoted as $p_B \in S_B$) is thus the closet point on $S_B$ to $p_{A \to B}$.

**Confidence of Correspondence.** For each point and its correspondence pair (e.g. $(p_A, p_B)$), we can also compute a confidence score $C(p_A, p_B)$ to measure the confidence of this mapping:

$$C(p_A, p_B) = 1 - D(p_A, p_B) \tag{4}$$

where $D$ refers to the normalized Euclidean distance between $p_A$ and $p_B$ in the 3D world coordinate. Similar to [12], if $C$ is lower than a pre-defined threshold $\tau$, we conclude that point $p_A$ does not have a correspondence on shape $S_B$.

## 4 Experiments

In this section, we present evaluations of the proposed dense correspondence learning approach. To the best of our knowledge, there is no benchmark that provides ground-truth dense correspondences for general objects, which is exactly our motivation to learn dense correspondence with self-supervision. However, thanks to the learned dense correspondences across pairs of instances, we are able to carry out the task of 3D semantic keypoint transfer and part segmentation label transfer to evaluate the proposed method as in [12]. In the following, we first introduce our experimental setup as well as baselines. We then report quantitative and qualitative comparisons with these baselines for 3D semantic keypoint transfer and part segmentation label transfer. We further present ablation studies to demonstrate the contribution of each component in the proposed model. Finally, we show that our model can generalized to real scanned point clouds unseen in the training stage. More results can be found in the appendix.

### 4.1 Experimental Setup

**Dataset.** We carry out the semantic keypoint transfer task on the KeypointNet dataset [46] as the BHCP benchmark used in Liu et al. [12] is not publicly available. Compared to the BHCP benchmark, the KeypointNet dataset is more challenging because: (a) it contains diverse objects and comes with large-scale annotations, (b) it is template-free and annotated by a large group of people, thus is less biased compared to the keypoints in the BHCP benchmark, which are from predefined templates. For the part segmentation label transfer task, we use the ShapeNet part dataset [47] as in [12]. For both datasets, we use the split provided in the original paper, and generate all pairs of shapes in the testing set as our testing pairs. To avoid interference from non-existing correspondences, we leave

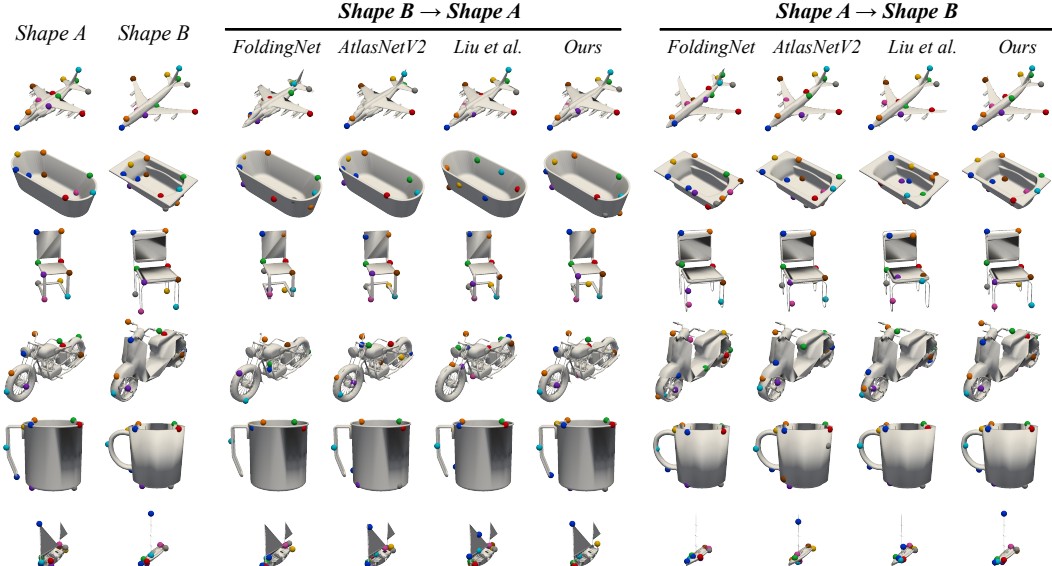

|  | Shape A | Shape B | Shape B → Shape A | | | | Shape A → Shape B | | | |
|---|---------|---------|------------|-----------|-----------|------|------------|-----------|-----------|------|
|  |  |  | FoldingNet | AtlasNetV2 | Liu et al. | Ours | FoldingNet | AtlasNetV2 | Liu et al. | Ours |

Figure 4: Keypoint transfer results for five categories: airplane, bathtub, chair, motorcycle, mug, and vessel. Each row contains two shape each with ground-truth keypoints and its pairwise transfer result.

|  | pla. | bag | cap | cha. | ear. | gui. | kni. | lam. | lap. | bik. | mug | pis. | roc. | ska. | tab. | avg. |
|---|------|-----|-----|------|------|------|------|------|------|------|-----|------|------|------|------|------|
| Liu et al. | 60.1 | 56.2 | 59.7 | 72.2 | 45.3 | **81.5** | 66.4 | 42.6 | 88.5 | 40.5 | **87.5** | **66.4** | 37.2 | 50.7 | 70.4 | 61.7 |
| Ours | **61.3** | **59.3** | **61.6** | **72.6** | **55.5** | 78.9 | **71.3** | **53.2** | **89.9** | **55.4** | 86.5 | 66.2 | **40.2** | **61.8** | **72.5** | **65.8** |

Table 1: Part label transfer results for 15 categories in the ShapeNet part dataset. Number measured with average IOU(%).

out instance pairs that do not share the same keypoint or part label. In all experiments, including our method and the baselines, we use a validation set for model selection.

**Baselines.** We evaluate the proposed method against state-of-the-art learning-based 3D dense correspondence prediction approaches, including AtlasNetV2 [18], FoldingNet [17], and Liu et al. [12]. Specifically, FoldingNet deforms from a fixed UV grid, while AtlasNetV2 explicitly allows the shape to be deformed from learnable elementary 3D structures. Neither of them estimates the confidence of correspondences. Liu et al. [12] propose a method that utilizes part features learned by the BAE-NET [11] to learn dense correspondence, including a mechanism that estimates the correspondence confidence.

**Implementation Details.** For both the point canonical mapping encoder and point inverse mapping decoder (see Figure 1), we follow [18] and use a three-layer MLP with ReLU activations, BatchNorm layers except for the last layer, where we use a hyperbolic tangent activation to obtain the final output. The training phase of our approach consists of two stages: (1) A pre-training stage trained with $L_{ACD}$ (Eq. 1) and $L_{rec}$ (Eq. 2) using $\alpha = 1$ for $L_{ACD}$ (2) A fine-tuning stage trained with $L_{ACD}$, $L_{rec}$, and $L_{cross}$ (Eq. 3) where we set $\alpha = 0$ for $L_{ACD}$. The total loss is formulated as $L_{total} = \omega_1 L_{ACD} + \omega2 L_{rec} + \omega_3 L_{cross}$, where $\omega_1 = 10$ for the pre-training stage; 20 for fine-tuning stage, $\omega_2 = 1, \omega_3 = 10$. For all experiments, we set $k = 2048, \tau = 0.9$ (see Section 3.2), and the parameters of the network are optimized using the Adam [48] optimizer, with a constant learning rate of $1e^{-4}$.

## 4.2 3D Semantic Keypoints Transfer

For fair comparisons, we follow [12, 37] and compute the distances from transferred keypoints to ground truth keypoints and report the percentage of testing pairs where the distances are below a given threshold in Figure 3. We demonstrate that for 11 out of 12 categories (e.g. airplane, chair, mug, etc.), keypoints transferred via our learned correspondence are more accurate than other methods [12, 17, 18]. At the distance threshold of 0.05, our method performs 11.2% more accurately than Liu et al. [12] on average of all categories. Figure 3 demonstrates the qualitative results of the keypoint transfer task. Even for categories with large intra-class variation, e.g. bathtub, motorcycle, or vessel, our method is able to transfer keypoints accurately thanks to the learned dense correspondence.

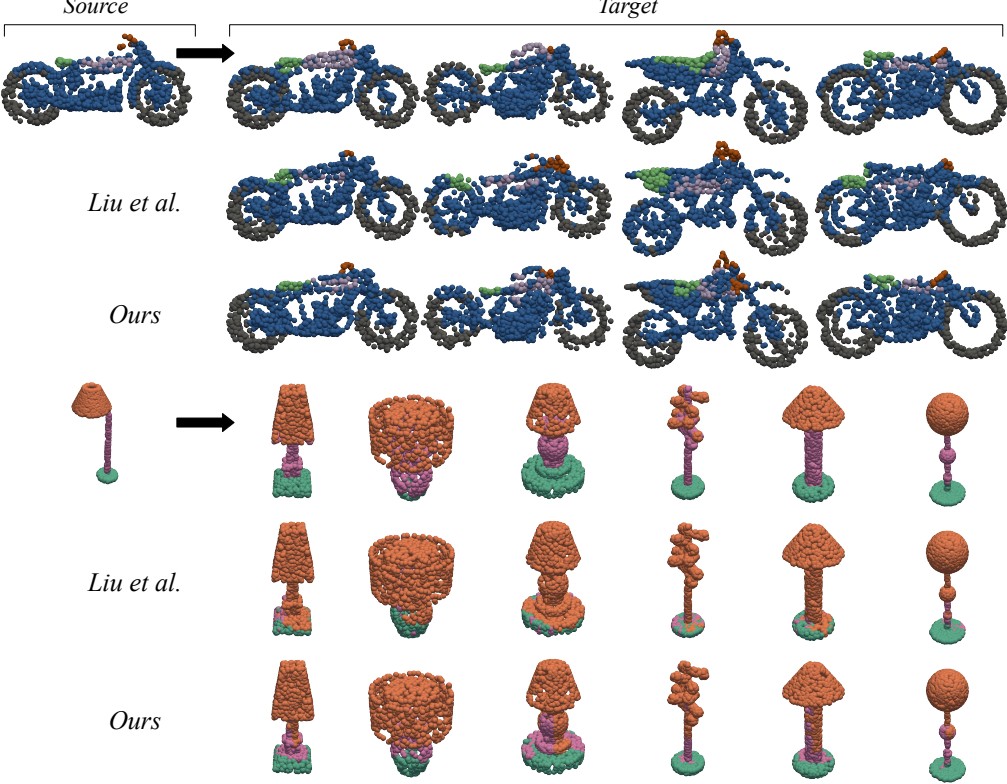

Figure 5: Qualitative results of part label transfer. The 1st row of the target indicates the ground truth instance part labels, while the rest shows the label transferring results via learned correspondences.

## 4.3   Part Segmentation Label Transfer

We further validate our approach on the part label transfer task and present quantitative results in Table 1 and qualitative results in Figure 5. Note that the settings are different from that of Liu et al. [12], which directly utilizes the branched co-segmentation results for evaluation (thus different quantitative results between the Table 1 and that in their paper). In Table 1, we show the intersection-over-union (IoU) between transferred and ground truth part labels. Our method performs better than Liu et al. [12] in 12 out of 15 categories and has a higher average IoU. For categories with large intra-class variations, such as lamps, motorbikes, our approach significantly outperforms Liu et al. [12]. The performance difference is most likely due to the branched co-segmentation [11] adopted by Liu et al. [12]. Branched co-segmentation relies heavily on the size of training data available. Such an approach is vulnerable to large shape variations [11], and is unable to segment flat surfaces. Moreover, its performance is sensitive to the pre-defined number of branches. Instead, our method naturally links different instances through a canonical primitive and is thus more robust to large shape variations. We show the qualitative results in Figure 5. Thanks to the dense correspondence learned by the proposed method, we are able to transfer small parts more accurately such as the seat of a motorcycle (the green points) and the pipe of a lamp (the purple points) comparing to Liu et al. [12].

## 4.4   Correspondence Confidence

Given a source shape $S_A$ and a target shape $S_B$, for every point $q \in S_B$, our model computes a confidence score indicating whether its corresponding point exists in $S_A$, as discussed in Section 3.2. As annotations can be biased by the annotator's definitions on keypoints, there is no absolute ground-truth for non-existence label between a shape pair. Thus, we visualize the confidence score as heatmaps for multiple target shapes in the airplane category in Figure 6 and compare with Liu et al. [12]. As shown in Figure 6, the proposed method is able to produce a more fine-grained confidence score compared to Liu et al. [12]. This is because our approach explicitly evaluates the confidence of

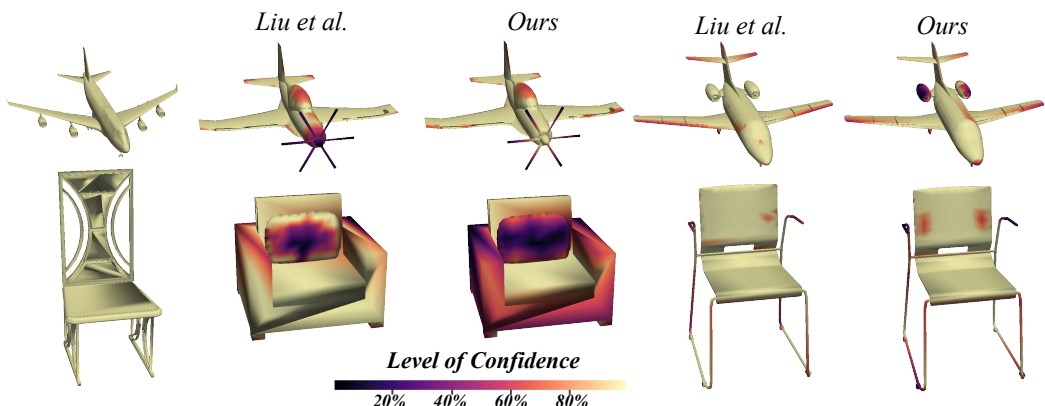

Figure 6: Heatmaps representing the correspondence confidence generated by our network. Source shape is in the leftmost of each row and dark color in each heatmap refers to low confidence in existing correspondence.

correspondences at a more fine-grained level – the distance between points, instead of a distance at the part-level, as proposed in [12].

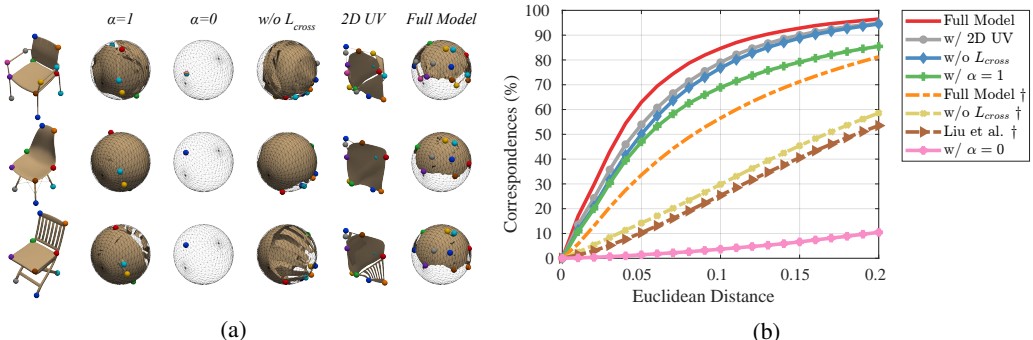

Figure 7: Qualitative (a) and quantitative (b) results of ablation studies on (i) the cross-reconstruction loss term (ii) the adaptive Chamfer loss term (iii) different types of canonical UV primitive using the chair category in the KeypointNet dataset. Note that the † sign indicates methods trained with an un-aligned setting.

## 4.5 Ablation Studies

**Effectiveness of cross-reconstruction.** The cross-reconstruction architecture is designed to ensure that corresponding points overlapping as much as possible on the canonical space. In our experiments, we find that with the datasets containing shapes with aligned 6D poses [10, 46], a single branch encoder-decoder framework (i.e., w/o $L_{cross}$) already produces a reasonable prediction of correspondences. However, as shown in Figure 7 (red vs blue), our cross-reconstruction framework significantly improves the performance with a large margin.

In addition, we validate the effectiveness of the framework on un-aligned shapes, by rotating the input point cloud with radian noise $\mathcal{N}(0, 0.5^2)$. We note that all co-segmentation approaches assume that the pose of input shapes are consistently aligned [11, 12]. Consistent with the assumption, such rotation severely degrades the performance of Liu et al. [12] (brown line). In contrast, with the help of cross-reconstruction loss (orange line), our model is rotation-invariant to a certain degree.

**Effectiveness of adaptive Chamfer loss.** To show that the adaptive Chamfer loss is necessary, we train our model with constant $\alpha = 0$ and $\alpha = 1$, respectively. The results are shown in Figure 7. When $\alpha$ is set to zero consistently, the primitive would eventually condense to a single point in the canonical space, therefore, significantly hurts the performance (pink line). On the other hand, if $\alpha = 1$, it is equivalent to enforcing a single canonical primitive for all instances by encouraging the

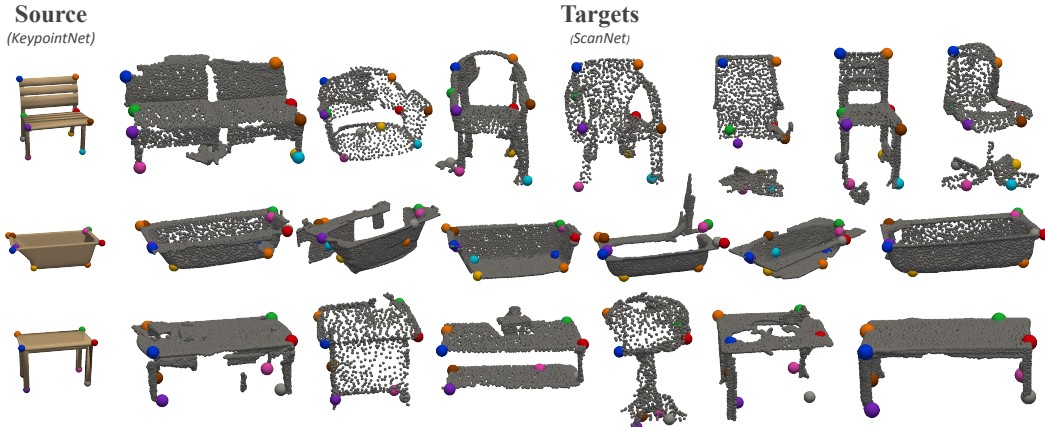

Figure 8: Keypoint transfer results from synthesized data to real scanned objects. We show results on the chair, bathtub, and table category – ScanNet categories that intersect with the KeypointNet dataset. Source shape is in the leftmost of each row.

primitive to cover the entire canonical space. This ignores the intra-class variation and deteriorates the performance as shown in Figure 7 (green line).

**Using a 2D UV grid vs. 3D UV sphere as a primitive.** To analyze the effect of different canonical UV primitives, we further train our model using a 2D UV grid instead of a 3D UV sphere. Figure 7 shows that parameterizing the canonical space with a 3D sphere quantitatively outperforms a 2D grid (grey line). The main reason is that the sphere is continuous at any point while a 2D grid is discontinuous at boundaries.

### 4.6 Testing with Real Scanned Data

To demonstrate that our model is robust to the domain gap between real and synthesized 3D objects, we apply our model to the real scanned objects. Specifically, we train our network on the KeypointNet dataset and test the trained model on real scanned point clouds from the ScanNet dataset [49]. As the ground truth keypoints or semantic part annotations in ScanNet are unavailable, we show the qualitative results in Figure 8. Although our model is only trained using synthesized data, it successfully predicts the correspondence and transfers keypoints from the source to the target shape.

## 5 Conclusions

In this work, we propose a self-supervised model, CPAE, that learns dense correspondence between 3D shapes in the same category. We introduce a canonical UV sphere, where dense correspondence for all the shapes can be explicitly obtained from. The key is to learn a 3D world coordinate to canonical space mapping, so that points from different instances are regarded as a pair of correspondences if they overlap on the sphere. We fulfill it through an autoencoder equipped with an adaptive Chamfer loss in the bottleneck, and a cross-reconstruction structure in the decoder. Experimental results validate the proposed method performs favorably against state-of-the-art schemes in various tasks and ablations. We show that our model is much more robust to rotation than the existing approaches that are non-rotation invariant by design. The task of learning rotation-invariant representations for 3d dense correspondence is challenging and still far from being solved. In future, we plan to extend our work to handling larger rotations.

## Acknowledgements

This work was supported in part by the MOST, Taiwan under Grants 110-2634-F-007-016, MOST Joint Research Center for AI Technology, All Vista Healthcare, and NSF CAREER grant 1149783. We thank National Center for High-performance Computing (NCHC) for providing computational and storage resources.

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
