# OpenReview forum: "Learning 3D Dense Correspondence via Canonical Point Autoencoder"
_NeurIPS.cc/2021/Conference — NeurIPS 2021 Poster_

### Official Review · Reviewer_DYH7 · 2021-07-12

**Rating:** 6
**Confidence:** 4

**Summary:**

The paper proposes an autoencoder architecture that decouples point cloud’s representation into the global latent vector and spherical canonical embedding (in $\mathbb{S}^2$ space) for each of its points. It is trained in the unsupervised way on the category-specific point clouds. This canonical map turns out to be robust to missing parts due to a specially crafted regulariser. What’s new related to the prior art is that the method is somewhat robust to initial orientation of the shape. The formulation is simple and elegant, however, the method is tested only on synthetic ShapeNet data, so it is unclear whether it would work well in a more practical setting. There are also other methodological concerns and clarity problems.

**Ethical Concerns:**

Re 4b in the checklist: strictly speaking, you not legally allowed to use data if you don’t have a license for it; I’d recommend ask the owners to issue you a license.

**Limitations And Societal Impact:**

There is no dedicated limitation section. What should be discussed:
* applicability to real-world data;
* the paper is over-claiming that it is robust to misaligned input (e.g. line 10); in fact, the paper experiments only with a limited angle variation and the accuracy still degrades quite a bit;
* the paragraph starting with line 189 claims that the shape can be generated by sampling the sphere; however some regions of the sphere can be optional, e.g. a bicycle can have either disk brakes or V-brakes, while the generated point cloud will have both, which is not realistic.

**Main Review:**

Significance / Experiments

(+) the ability to ignore parts of $\mathbb{S}^2$ embedding space with asymmetric Chamfer regulariser is novel and useful;

(+) most methods in this area train on ShapeNet with aligned cameras; such a setting would be difficult to achieve with the real data; this method instead claims to be somewhat invariant to rotations: if this useful property generalises to real data, it can be used for global rigid alignment, e.g. to initialise ICP or to fit a parametric model;

(−) experiments on rotation invariance are somewhat limited though: rotations are sampled from Normal distribution with STD=0.5 radians, which is less than 30˚ , so it hardly tests the ability to perform global alignment; a proper test would be to take random rotations; even given that limited setup, the method is not very accurate: from Fig. 7b, at the distance = 0.05, it gets 30% correspondences correctly vs 60% of the aligned case;

(−) all experiments are performed on the synthetic ShapeNet dataset; there can be a significant domain gap with the real data that would be obtained by a LiDAR or dense stereo;

(−) the performance is measured on sparse keypoints, which are usually defined in distinctive points; this measurement can be biased; I suggest to evaluate the method on datasets of SMPL human body models in the variety of poses, where the correspondences are given by mesh structure.

Methodology

(=) the 3D point coordinates are concatenated to the 512D latent code to be fed into the MLP; I would expect the contribution of the first part to be saturated; would some other way of conditioning be more reasonable?

(−) in eq. (2), why are the CD and EMD terms needed? When MSE is satisfied, the other two should be as well;

(−) line 253: do I understand correctly that tanh is used in the end of the MLP that generates points? Its output is bounded to (−1, 1) then; what if the original point cloud would not fit a unit cube? Regardless, it seems like a strange inductive bias to include in the model. (Generating points in the Fourier space might be a good idea)

Originality

(+) asymmetric Chamfer distance loss to the unit sphere is a new idea which appears to be crucial for being able to learn the spherical embedding;

(−) on the other hand, I don’t see why the first term in ACD, which makes the points to lie on the unit sphere, is needed; why not just normalise the embeddings? (I have a hypothesis but would be good to confirm that it does not work) The weight $\alpha$ will then balance half of the CD vs MSE. Also, can the second term be replaced by entropy to avoid sampling/discretisation artefacts? Comparing to those simpler baselines would shed the light on the main contribution.

(=) all other components have been used in point cloud autoencoders (including cross-reconstruction loss and PointNet encoder).

Clarity

(−) The paper provides loss formulations (1–3) but does not say how they are combined, e.g. with what weights;

(−) is $\alpha$ *gradually* decreased during training (line 151) or in one step (line 255)?

(−) I don’t understand the theoretical argument in Section 3.2. Also, in eq. (5), I don’t understand the idea of using Euclidean distance in the world space to determine the confidence? What if $S_A$ and $S_B$ are rotated w.r.t. each other?

Notes to the authors (no effect on rating):
* I would not call such method self-supervised; the terminology is vague here but I think autoencoders that match the output to the input are unsupervised methods (self-supervision usually assumes some modification of the input for supervision);
* line 48: the sentence is not in grammatical agreement;
* line 89: the algorithm is feasible to calculate ← is able to calculate;
* line 146 refers to the pink box in Fig. 1; I don’t see it;
* line 159: MES loss;
* line 161: 1e1 ← 10;
* line 165: a Chamfer loss ← Chamfer loss;
* line 310: equivalent to enforce ← equivalent to enforcing.

=========

UPD. Thank you for engaging in a discussion and diligently evaluating new baselines. My main concerns are now resolved, to an extent. For rotation invariance the method indeed beats the weak baseline (non-rotation invariant method by design). Please follow-up on the promises to re-position the paper, i.e. claim only invariance within small angles and acknowledge that the problem is far from being solved even for small angles. The new experiments on the ShapeNet crops and nearest-neighbour baseline are steps in the right direction; I encourage you to find ways to evaluate quantitatively. The clarity will hopefully be sorted out with the help of our comments.

**Time Spent Reviewing:**

4 hours

---

> ### Author Response · Authors · 2021-08-10
> **Response to Reviewer DYH7**
>
> 1. **RE: Experiments on rotation invariant are somewhat limited**
>
>     We conduct further experiments on verifying the robustness of our model against unaligned point clouds. Please see our general response section (A) for details.
>
> 2. **RE: No experiments on real data**
>
>     To the best of our knowledge, there are no existing labeled large-scale datasets for man-made objects using real scans. Therefore we follow prior work [14] and use synthetic data to evaluate our approach. Moreover, the domain gap between real-world and synthetic point clouds (i.e., XYZ locations) is relatively small compared to that of image datasets (i.e., RGB pixels). Such domain gap is often introduced by the imperfect dataset capturing process (imperfect scanning) and can be alleviated by various preprocessing methods (e.g. shape completion and denoising), which is not a focus of this work. However, as suggested by the reviewer, we provide experimental results on real scan data using the SMPL human model dataset in the following response.
>
> 3. **RE: No experiments on dense task/Suggestion on evaluation using  SMPL human model dataset**
>
>     We not only evaluate our method on sparse keypoints but also dense part labels. We argue that learning dense correspondence for human body models is a very different task from which we aim to solve.  Human models are all genus-zero shapes and therefore can be fit with parameterized primitives, such as a carefully designed mesh template. In addition to that, human pose articulations can be learned with priors such as geodesic distance consistency as they are approximate isometries.  In this work, we focus on man-made objects (as in [14]) that have none of the aforementioned characteristics.
>
>     As suggested, we conduct experiments on the real scan data in the FAUST humans dataset, which is a high-resolution benchmark that includes noise and has holes.  Our setup follows [16] and quantitative results were evaluated through the online evaluation system of the inter-subject challenge track. We compared our method with a supervised [15] and an unsupervised [16] approach and reported the average error in the table below.  Note that for fair comparisons, we use the variant reported in [16] without isometric regularization.
>
>     |                                  | Average Error (lower is better) |
>     |---|---:|
>     | FMNet [15] (supervised)            |          4.83 |
>     | Ours                             |          6.78 |
>     | Groueix et al. [16] (unsupervised) |          8.72 |
>
>
>
> 4. **RE: Does the concatenation of the 3D point coordinate and 512D latent code saturate the location information? Would some other way of conditioning be more reasonable?**
>
>     No. Suppose the 512D latent code dominates and the network ignores the coordinates, then all the points share the same input, i.e., the replicated latent code, and all the outputs should be identical $-$ this contradicts our results.
>
>     We emphasize that concatenating the latent code and coordinates is an effective way to fuse the two information streams. On one hand, if the location information is missing, the predicted coordinate for each point would be identical. On the other hand, if the shape information in the latent code is missing, we would not reconstruct different instances. Searching for a more powerful neural architecture for the conditional implicit function is a very interesting direction. However, although we have not yet explored this direction, our current architecture follows several state-of-the-art prior works [17,18,19] where all of them use concatenation to condition their implicit function on a latent code.
>
> 5. **RE: Why are both CD and EMD terms needed in eq. (2)? When MSE is satisfied, the other two should be as well**
>
>     This is to impose a smoothness constraint on the reconstructed surface, as MSE loss alone does not contain any local geometry information. When a point fails to satisfy MSE, the other two terms can provide supervisory signals to guide the transformation.
>
> 6. **RE: Is the activation function Tanh used at the end of the MLP? Why?**
>
>     That is correct. We follow the data preprocessing procedure in [17,18] where the input point cloud is normalized into a unit sphere. Therefore, it is guaranteed that all input samples are within the range of (−1, 1). In that case, it is fairly reasonable to use Tanh to bound the output.
>
> 7. **RE: Suggestion on adding more simple baselines (using norm to replace the first term in $L_{ACD}$/using entropy to replace the second term in $L_{ACD}$)**
>
>     As suggested, we add a baseline on replacing the first term in $L_{ACD}$ with an L2 norm. This results in an average distance of 0.0846, which is worse than 0.0541 when our full model is used. We found that models trained with the L2 norm are likely to simply expand and stick the input to the sphere surface. Although the L2 norm encourages the embedding to lie on the unit sphere, as it does not contain nearest neighbor information as $L_{ACD}$ does, it fails to preserve the input shape information and learn a meaningful embedding.
>
>     The reviewer also suggests a baseline using entropy to replace the second term in $L_{ACD}$. However, as our approach does not model point clouds as probability distributions, it is not possible to directly compute the Shannon entropy. One possible workaround is to estimate the kernel density function. Nevertheless, this is considerably more complicated than our proposed method and is beyond the scope of our paper.
>
> 8. **RE: How are the loss terms combined?**
>
>     The total loss is formulated as $L_{total} = \omega_{1} L_{ACD} + \omega_{2} L_{rec} + \omega_{3} L_{cross}$, where $\omega_{1}=10$ for stage1; $20$ for stage2, $\omega_{2}=1, \omega_{3}=10$. These weights are chosen empirically to balance the loss terms into similar scales.
>
> 9. **RE: Is $\alpha$ gradually decreased during training or in one step?**
>
>     We clarify this in the general response section (D). We will clarify that in our revised version.
>
> 10. **RE: Clarify the theoretical argument in Section 3.2**
>
>     Please see the general response section (C) for clarification.
>
> 11. **RE: The intuition of using Euclidean distance in world space to determine confidence. What if the inputs are rotated?**
>
>     Please see the general response section (A) for clarification.
>
> 12. **RE: Paragraph starting with line 189 claims that the shape can be generated by sampling the sphere; however as some regions of the sphere can be optional, the generated point cloud will be unrealistic**
>
>     Here our emphasis is that, as a merit of implicit function, our mapping functions can learn continuous mappings. Therefore, one can interpolate points to reconstruct a nearly continuous surface. Instead of sampling the sphere randomly, here we mean interpolation can be conducted within $U_{A}$ to produce a continuous surface. We will correct the confusion in the revised paper.  We also note that neither reconstruction nor generation is the goal of our paper.
>
> 13. **RE: Over-claim in abstract**
>
>     We are aware that we missed the term “within a certain rotation range" (as mentioned in L57 and L305) in our abstract. We will revise that accordingly.
>
> 14. **RE: Ethical concern on data license**
>
>     We have acquired permission and license from the authors in both KeypointNet and ShapeNet Part dataset.
>
>
> ---
>
> *[14] Feng Liu and Xiaoming Liu. Learning implicit functions for topology-varying dense 3d shape correspondence. Advances in Neural Information Processing Systems (NeurIPS), 2020.*
>
> *[15] Or Litany and Tal Remez and Emanuele Rodolà and Alex M. Bronstein and Michael M. Bronstein. Deep Functional Maps: Structured Prediction for Dense Shape Correspondence. IEEE International Conference on Computer Vision (ICCV), 2017.*
>
> *[16] Thibault Groueix and Matthew Fisher and Vladimir G. Kim and Bryan C. Russell and Mathieu Aubry. 3D-CODED : 3D Correspondences by Deep Deformation. European Conference on Computer Vision (ECCV), 2018.*
>
> *[17] Yaoqing Yang and Chen Feng and Yiru Shen and  Dong Tian. FoldingNet: Point Cloud Auto-encoder via Deep Grid Deformation. IEEE Conference on Computer Vision and Pattern Recognition (CVPR), 2018.*
>
> *[18] Thibault Groueix and Matthew Fisher and Vladimir G. Kim and Bryan C. Russell and Mathieu Aubry. AtlasNet: A Papier-Mâché Approach to Learning 3D Surface Generation. IEEE Conference on Computer Vision and Pattern Recognition (CVPR), 2019.*
>
> *[19] Alex Yu and Vickie Ye and Matthew Tancik and Angjoo Kanazawa. pixelNeRF: Neural Radiance Fields from One or Few Images.  IEEE Conference on Computer Vision and Pattern Recognition (CVPR), 2021.*

---

> > ### Comment · Reviewer_DYH7 · 2021-08-26
> > **Response**
> >
> > Thank you for the verbose response!
> >
> > My only substantial concerns about the paper now are lack of real-data experiments and the rotation invariance.
> > I respectfully disagree that the domain gap is small and can be resolved by engineering. E.g. as you mention, standard ShapeNet preprocessing fits objects into a unit sphere. With occlusions and outliers artefacts of the scanning process, this can lead to misestimation of scale, and the method will likely fail.
> >
> > Related to that, lack of full rotation invariance, which is a significant selling point of the paper. Once the shapes are globally aligned, it is relatively easy to guess the keypoint locations. From the new results, the errors are quite sensitive to misalignment. I acknowledge that the numbers improve over the baseline that does not attempt to relax this assumption, but this won’t help much in analysing real scans where the rigid alignment may not even be uniquely defined, let alone easily detected.
> >
> > I think lack of the public datasets does not automatically mean that the paper makes progress towards solving the real task. This should still be demonstrated in some way, e.g. in transfer-learning setting from ShapeNet to real scans. I also think Objectron dataset was available by the time of submission.
> >
> > Overall, the paper makes a good contribution but these two problems drag it down quite a bit.

---

> > > ### Author Response · Authors · 2021-08-28
> > > **Response to Reviewer DYH7**
> > >
> > > We thank reviewer DYH7 for the response and would like to address two issues recently raised:
> > >
> > > * Our model is robust to the domain gap between real and synthesized 3D objects. Following the NeurIPS policy$^{\dagger}$, we show qualitative keypoint transfer results$^{\ddagger}$ on this anonymous webpage (https://imgur.com/a/BiefOJx) by applying our model to the real scanned objects from the ScanNet [1]. Although our model is only trained using synthesized data from ShapeNet, it successfully predicts the correspondence and transfers keypoints from the source to the target shape. As the ground truth keypoints or semantic part annotations in ScanNet are unavailable, we are not able to quantitatively evaluate real data. We will include these qualitative results and discussions in the revised manuscript and supplementary material.
> > > * For rotation invariance, we would like to emphasize that:
> > >     1. Our claim in the paper (Line 58) -- “our model is rotation-invariant **within a certain rotation range**.“ is not “full rotation invariance” or “a significant selling point of the paper”, as reviewer DYH7 has commented. Instead, we aim to **relax the strict alignment requirement** in existing SOTA works [2].  In “General Response (A)” in the rebuttal, we have demonstrated that, quantitatively, our method outperforms  SOTA methods in the unaligned setting. We agree that both existing and our method cannot solve the unaligned setting perfectly, but our paper is the first attempt towards that direction. The fact that we are able to deliver superior quantitative results justifies our contribution and can be a starting point for future research along this line.
> > >     2. We also show that even in the aligned settings, precisely transferring keypoints is not a trivial task. To this end, we introduce a baseline that uses the nearest neighbor point as correspondence in two aligned point clouds. Specifically, for each keypoint in the source point cloud, we find its nearest neighbor point in the aligned target point cloud as its correspondence.  The results are provided in this anonymous link (https://imgur.com/a/eohaNtm), where we plot the correspondence accuracy (Y-axis) when an error is below a given threshold of Euclidean distance (X-axis). At the threshold of 0.05, the baseline gets only 38% accuracy (green line) compared to 62% of our approach (blue line). On top of that, our approach consistently outperforms the baseline (green line, aligned setting) when input point clouds are rotated with $\pm$30$^{\circ}$ (orange line).
> > >
> > >
> > > > $\dagger$ The NeurIPS submission policy (https://neurips.cc/Conferences/2021/PaperInformation/NeurIPS-FAQ) states *“May I include a link in the author response?  External links are discouraged, but you may include a link if it is required to respond to a question from a reviewer.”*.  We follow the policy and include this anonymous link to show qualitative results to answer reviewer DYH7’s questions.
> > >
> > > > $\ddagger$ We show results on the chair, table, and bathtub category -- ScanNet categories that intersect with the KeypointNet dataset.
> > >
> > > *[1] Angela Dai and  Angel X. Chang and  Manolis Savva and  Maciej Halber and  Thomas Funkhouser and  Matthias Nießner. ScanNet: Richly-annotated 3D Reconstructions of Indoor Scenes. CVPR, 2017*
> > > \
> > > *[2] Liu Feng and Xiaoming Liu. Learning implicit functions for topology-varying dense 3d shape correspondence. NeurIPS, 2020*

---

> > > > ### Author Response · Authors · 2021-08-31
> > > > **Please let us know whether you have additional questions**
> > > >
> > > > Dear Reviewer,
> > > >
> > > > We have provided more results and explanations based on your review. Please go over them and let us know whether you have additional questions or not.
> > > >
> > > > Thank you,

---

### Official Review · Reviewer_xuCY · 2021-07-16

**Rating:** 7
**Confidence:** 4

**Summary:**

The paper proposes a canonical point autoencoder for the purpose of unsupervised learning dense correspondences between 3D shapes of the same category. The canonical space is defined on a 3D sphere, but with extra flexibility that the template needs not to fully cover the full sphere. This gives the flexibility to generalize to non genus 0 shapes. The proposed method seems to give noticeable improvement over existing ones.

**Limitations And Societal Impact:**

Yes, I don't find any negative societal impact.

**Main Review:**

In general the idea of learning a point cloud autoencoder with the bottleneck defined as the projection onto a 3D sphere seems to be effective, as demonstrated by the authors in the experiments. The authors did a pretty good job in explaining their methods,  and the experiments are thorough.

Though I do have several concerns / comments for the paper.
(1) The intuition of why the proposed 3D sphere canonical space can handle non genus 0 shapes is provided very late in the paper, not until page 4 when “relax the bidirectional constraint of the Chamfer loss” is mentioned. The authors should give a clearer explanation about this in their intro, otherwise, a spherical canonical mapping is not enough detail to convince the readers that the proposed method is going to work for non genus 0 shapes. In regard to this, on page 3 Ln 99, “non-linear mappings” alone is too ambiguous and not enough to explain why the method is able to lift “such restriction”.

(2) In the intro, the author claimed the method works “.. even. when instances in the training dataset are not aligned, …, does not need to predict an additional rotation matrix ...”  I am not fully convinced by this claim. I fail to see why the proposed method is “rotational invariant”, and how the method is able to solve the ambiguity between shape variation and rotation. In addition, the way it computes confidence of correspondences are clearly not rotational invariant, as it directly evaluates the euclidean distance between two points in the 3D world coordinates. If the source & target point cloud differs by a significant amount of rotation, such confidence score would not make any sense.

(3) I find the “reconstruction of ordered point clouds” section on page 5 difficult to understand. I would suggest the author to make a more formal mathematical proposition. Moreover, the authors seem to try to prove something w.r.t. affine transformation, but why does that applies to MLP which is non-linear?


**Time Spent Reviewing:**

3

---

> ### Author Response · Authors · 2021-08-10
> **Response to Reviewer xuCY**
>
> 1. **RE: Intuition of why the proposed 3D sphere canonical space can handle non-genus-zero shapes.**
>
>     The term "relax the bidirectional constraint of the Chamfer loss" on page 4 refers to the loss term $L_{ACD}$, whose goal is to allow each instance to produce its own “instance primitive”. The reason why our approach can handle non-zero-genus shapes is independent of $L_{ACD}$.  Please see our general response section (B) for further clarification on the reason. We will give a clearer explanation in the revised version.
>
> 2. **RE: Intuition behind rotation invariant/Confidence score via Euclidean distance**
>
>     Our method is relatively robust to shape variation and rotation due to the use of $L_{cross}$ and the canonical space. For a detailed explanation of these two questions, please see the general response section (A).
>
> 3. **RE: Clarifying the “ Reconstruction of Ordered Point Clouds” section.**
>
>     Please see the general response section (C) for clarification.

---

### Official Review · Reviewer_yaub · 2021-07-20

**Rating:** 6
**Confidence:** 3

**Summary:**

The paper suggests an end-to-end learnable method to find dense 3D point correspondences of point clouds. The authors first project the input point clouds onto a canonical spherical UV space, where corresponding points of different shape instances map to similar coordinates. Then, an instance-based feature vector is concatenated with the UV coordinates and fed to a decoder to retrieve the original point cloud. Since the UV coordinates hold correspondence information, the output point cloud is, in fact, an ordered one.

**Limitations And Societal Impact:**

Limitations not discussed. A (possible) critical limitation of the model is the instability and the high sensitivity to the hyper-parameters. The authors should discuss these aspects in the paper.

**Main Review:**

Strengths:
The paper is well written. The problem setting is well-motivated, and the paper addresses an important research problem.

The model achieves state-of-the-art performance.

The purpose of each component of the model is well discussed.
The ablation study is insightful.

Concerns:

The biggest concern I have with the proposed approach is with the combination of loss components. I feel that the ideal combination of these different loss components might be too sensitive to the dataset bias, and the network might not be very stable. See below for detailed comments.

1) The whole methodology of this paper depends on the fact that corresponding points of the input point cloud instances (of the same class) get mapped onto similar UV coordinates. The loss which encourages this is $L_{cross}$. However, this loss is utilized using randomly chosen instances from the same class. What if the chosen random samples are part-missing etc. samples? If the authors specifically avoid such samples, they should mention this. I understand that other related works also do not address this problem. But this paper argues explicitly that finding different intra-class point correspondence (e.g., missing parts) is the main virtue of their work. Hence, I put more weight on this fact.

2) The authors claim that by reducing \alpha in $L_{ACD}$ gradually allows each shape to have its own UV coordinates to some "extent". Now, "extent" is a highly ambiguous term. For instance, $L_{ACD}$ can easily force the model to find different UV coordinates for similar instances also (not only part missing instances). There is no direct way to control this. Again, the authors can argue that $L_{cross}$ prevents this by forcing similar instances to have similar UV coordinates. But again, $L_{ACD}$ and $L_{cross}$ have different goals, and hence, I feel the model might be too sensitive to the competition of these loss components. Similarly, the model can be too sensitive and dataset dependent for the ideal gradual decreasing of \alpha. If alpha stays close to 1 too much, the model might not work well with part-missing instances. On the other hand, if \alpha stays close to 0 too much, even similar instances might have different UV coordinates.

3) Let's assume that by exhaustive parameter tuning, we find the ideal hyperparameters to avoid the above issues. Still, there is an inherent mismatch between the goals of $L_{ACD}$ and $L_{cross}$. The authors argue that somehow they can find an ideal point where the end-goal is satisfied. However, it is only guaranteed if the decoder is a linear transformation (Eq. 4). This is a very strong assumption that is clearly violated since the model uses a non-linear MLP. For example, a non-linear MLP can easily find a solution that satisfies both $L_{ACD}$ and $L_{cross}$ by projecting two similar instances onto different sets of UV coordinates and still reconstructing a similar output point cloud. Since the MLP is non-linear, it does not need to have similar UV coordinates to reconstruct similar outputs.

In summary, I feel that although the results are impressive, the model contains too many moving parts and hence is not a robust approach. However, I may be missing some crucial facts, and in that case, I invite the authors to enlighten me.

**Time Spent Reviewing:**

4

---

> ### Author Response · Authors · 2021-08-10
> **Response to Reviewer yaub**
>
> 1. **RE: What if the chosen random samples in $L_{cross}$ contain missing parts? Do we specifically avoid such cases?**
>
>     No, we do not specifically avoid such cases as we want to ensure no annotation is needed. The training of the network is robust as we use Chamfer loss for $L_{cross}$ to allow more tolerance on the cross reconstructed shapes. Compared to EMD distances that assume one-to-one bijective mappings, Chamfer loss has less restriction on the uniformity of the generated shapes.
>
> 2. **RE: The model can be too sensitive and dataset dependent for the ideal gradual decreasing of $\alpha$.**
>
>     As clarified in the general response section (D), the parameter $\alpha$ is implemented in a two-stage paradigm where $\alpha$ is simply set to 1 in the first stage and 0 in the second stage. There is no need to search for hyper-parameters for $\alpha$. For more details regarding the two-stage training paradigm, please see Algorithm 1 in our supplementary materials. We will clarify that in our revised version.
>
> 3. **RE: $L_{ACD}$ can force the model to find different UV coordinates for instances, therefore, compete with $L_{cross}$**
>
>     As stated in L48~49, $L_{ACD}$ maps input point clouds to the sphere, and $L_{cross}$ further aligns different point clouds on the sphere -- they serve for orthogonal purposes and thus do not compete with each other. We also observe the training curves for both drops smoothly. We will add them to the revised paper.
>
> 4. **RE: Even if $L_{ACD}$ and $L_{cross}$ no longer compete, a non-linear MLP can encode similar instances onto different sets of UV coordinates and still reconstruct a similar output point cloud.**
>
>     The extreme case where "two similar instances are projected onto different sets of UV coordinates and still reconstruct a similar output point cloud" is not likely to happen base on two reasons:
>     * As aforementioned, the end-goal can be achieved when the decoder is continuous, where MLPs do satisfy (as stated in the general question (C), we do not need the assumption of a linear transformation, which will be clarified in the paper).
>     * The overlapping of UV primitives is ensured due to the cross reconstruction framework, more specifically the $L_{cross}$. With the same latent code,  swapping $U_{A}$ and $U_{B}$ results in the same output shape $-$ which exactly enforces $U_{A}$ and $U_{B}$ to be aligned in the UV coordinates. This is even independent of what network architecture we use.
>
>     Again, we emphasize that the two losses serve for different goals as aforementioned where mismatch does not exist.
>
>
> 5. **RE: Exhaustive parameter tuning/Model instability/Robustness**
>
>     Our method is robust to hyper-parameters. We use the same set of hyper-parameters across all experiments, including 12 models (one for each category) in the keypoint transfer task and 16 models in the part label transfer task.

---

### Official Review · Reviewer_kS8b · 2021-07-23

**Rating:** 6
**Confidence:** 4

**Summary:**

This paper presents a novel solution for learning dense correspondence between un-aligned 3D shapes in an self-supervised manner. The key contributions this paper is to design a novel canonical space for representing all 3D shapes and learning an auto-encoder to learn a mapping to and from an input point cloud to this canonical space. The effectiveness of this approach is shown for the 3D semantic keypoint transfer and part segmentation transfer tasks across different categories.

**Ethical Concerns:**

They havent been discussed.

**Limitations And Societal Impact:**

The authors do mention certain limitations of their approach. However, it would be good if they discussed why intuition behind why some of the categories do not perform well for part segmentation label transfer task.

**Main Review:**

## Strengths

1. The paper proposes a novel canonical space for representing 3D shape, which helps to learn correspondence of shapes.
2. The paper proposes a way to learn an auto-encoder, which can learn how to map to and from the input point cloud and the canonical space. Besides the loss functions, used in [9], they use a modified version of Chamfer loss at the canonical representation space as well as a cross reconstruction loss where the encoded features from a particular shape instance is transfered to another shape instance.
3. The paper shows the effectivess of this approach on the 3D semantic keypoint transfer and part segmentation transfer tasks where they show results across different categories.

## Weaknesses
1. **Use of sphere as the canonical representation**: There are several aspects that are unclear about using sphere as the canonical representation
  a. *Motivation*: The paper does not talk about the motivation behind using sphere as a primitive for the canonical representation. Shouldn't projecting a 3D shape onto the surface of the sphere lose the convexity and concavity of the different 3D shapes? The authors should also cite papers from the rich literatures (including recent literature on 3D shape reconstruction) that has explored different kinds of primitives.
  b. *Empirical Results* From the emprical perspective, the fact that the yellow (w/o $ L_{cross}^{\dagger} $) and brown (Liu et al) lines are close by in Fig 7(b),  show that this just using this representation is not significantly useful. Moreover, Fig 9 shows that $ \alpha = 0 $, does not work at all where the representation in the primitive space collapses to a single point. Isn't this an indication that this representation is not powerful enough to capture a good canonical representation?
 c. *Loss at the encoded representation layer*: Why is Chamfer loss used at the encoder space? This is not supposed to be exact input 3D shape. What is the motivation of using such a loss function instead of loss functions which only looks at local surfaces or neighborhoods? Some recent papers that come to mind are as follows:
    - Smirnov, Dmitriy, et al. "Deep parametric shape predictions using distance fields." CVPR. 2020.
    - Sharma, Gopal, et al. "Parsenet: A parametric surface fitting network for 3d point clouds." ECCV 2020.
 d. It is not clear why it is expected that the rare components would appear far away from the primitive. Is this infered emprically and/or is there an high level intuition why this would happen?

2. **Training details**: It would be good to mention more details about the training approach:
  a. *Training data*: Which dataset do you train on? How many categories are there? How many shape instances are there per category?
  b. *Cross-reconstruction loss*: It is not clear whether the other point cloud is sampled randomly from the same category or from a different category. During training, in a min-batch, how is the batch setup from this perspective?
  c. Do you perform data augmentation on the input point cloud, especially for the cross representation loss part? If no, why would this not help?
  d. Why do you rotate by only sampling from $ \mathcal{N} (0, 0.5^2) $? Is that parameter chosen randomly or there is some empirical justification?


3. **Relation to cycle consistency**: The cross-reconstruction loss has similarities to cycle consistency. This is definitely a novel formulation for this task. But it would be interesting to cite related works on this topic and discuss what are the differences of this loss compared to others. Somne of the papers related on this part are:
  i. Kulkarni, Nilesh, Abhinav Gupta, and Shubham Tulsiani. "Canonical surface mapping via geometric cycle consistency." ICCV 2019.
  ii. Zhou, Tinghui, et al. "Learning dense correspondence via 3d-guided cycle consistency." CVPR 2016
  iii. Aumentado-Armstrong, Tristan, et al. "Cycle-Consistent Generative Rendering for 2D-3D Modality Translation." 3DV 2020.
  iv. Groueix, Thibault, et al. "Unsupervised cycle‐consistent deformation for shape matching." Eurographics 2019.








**Time Spent Reviewing:**

~5

---

> ### Author Response · Authors · 2021-08-10
> **Response to Reviewer kS8b**
>
> 1. **RE: Use of sphere as the canonical representation**
>
>     **(a)** As clarified in the general response section (B), our approach is not limited to genus 0 shapes and can therefore preserve the convexity/concavity in different 3D shapes. We will make this more clear in the revised version.
>
>     **(b)** We note that the yellow (w/o $L_{cross}$) is with the ***unaligned setting*** (noted with $\dagger$), in which the ***$L_{cross}$ is critical***. W/o $L_{cross}$, it is reasonable to see the performance drops as low as the Liu et al. Our Full Model$\dagger$ that having the $L_{cross}$ shows significantly better performance (the orange line).
>
>     We note that setting $\alpha$ constantly as 0 (Fig. 7a) won’t work: Chamfer distance needs to be bidirectional, where the second term exactly encourages the mapped point to be scattered uniformly to the sphere, instead of collapsing to the same point.
>
>     **(c)** The motivation to use global Chamfer loss is that we need to enforce all mapped points to scatter on the surface of a sphere $-$ it is a set-to-set alignment that requires a global operation. In our case, there is no shape prior to the output of the encoder, thus there is no way to perform shape decomposition or primitive fitting in order to use local loss functions.
>
>     While local losses are effective for the suggested works, we note that the shape matching procedures in these works are very different from ours. E.g., both Sharma et al. [6] and Smirnov et al. [5] aim at fitting patches or cuboids to the actual shapes, which vary significantly to ours that try to encode shapes to a canonical space. In addition, [6] does not infer any correspondence with unordered patches, and [5] can only infer part-level correspondences and requires additional supervision such as the number of primitives. We will add and discuss them in the revised paper.
>
>     **(d)** Yes, we infer that the rare components would appear far away from the primitive empirically. From a data-driven perspective, these components are minorities in the dataset and it's hard for them to occupy a distinct area in the canonical space.
>
> 2. **RE: Training details**
>
>     **(a)** We evaluate the keypoint transfer task on KeypointNet dataset on 12 categories including airplane(1022), bathtub(492), cap(38), car(1002), chair(999), guitar(697), helmet(90), laptop(439), motorcycle(298), mug(186), table(1124), and vessel(910). For the part segmentation label transfer task, we use the 16 categories in ShapeNet part dataset, which includes airplane (4027), earphone(73), cap(56), motorbike(336), bag(83), mug(213), laptop(452), table(8420), guitar(793), knife(420), rocket(85), lamp(2308), chair(6742), pistol(307), car(7496), and skateboard(152).
>
>     Note that the number in the parenthesis refers to the number of shape instances per category.
>
>     **(b)** We use samples from the same category to obtain $L_{cross}$. For efficiency purposes, we do not explicitly generate all possible combinations in the training set. Instead, we shuffle the dataset every epoch and generate pairs inside a mini-batch.
>
>     **(c)** We do follow regular data augmentation procedures used in [7], where Gaussian noise is added to the input point cloud. Aside from that, we apply random rotation augmentation as in [8] for the unaligned setting.
>
>     **(d)** Please see our general response section (A) where we provide empirical justifications.
>
> 3. **RE: Relation to cycle consistency**
>
>     Indeed, our cross reconstruction loss is related to literature that uses cycle consistency as a supervisory signal to learn correspondence without ground truth annotations. Some of these works exploit the consistency among different modalities (e.g., 2D-3D).  Specifically, Zhou et al. [10] aim to learn cross-instance pairwise correspondence for 2D images using cycle-consistency terms guided by 3D synthetic data. Kulkarni et al. [9] and Aumentado-Armstrong et al. [11] both leverage cycle-consistency terms to enforce correspondence between a 2D image and a 3D canonical template. Aside from these works that focus on cross-modality, work from Groueix et al. [12] is mostly related to ours. They propose a “point-wise” cycle consistency loss that explicitly enforces a point deformed through any cycle of deformations to be mapped back to the origin location. Different from their approach, our consistency is instance-wise and we do not assume bijection mapping among shapes. We will cite and discuss these related works in our revised version.
>
> 4. **RE: Limitations on part segmentation label transfer task**
>
>     We are aware that some categories do not perform well for the part segmentation label transfer task. We suspect one reason is that our approach does not generate smooth correspondence boundaries at the joint of connected parts. We will discuss this in our revised paper.
>
> ---
>
> *[5] Dmitriy Smirnov and Matthew Fisher and Vladimir G. Kim and Richard Zhang and Justin Solomon. Deep parametric shape predictions using distance fields. IEEE Conference on Computer Vision and Pattern Recognition (CVPR), 2020.*
>
> *[6] Gopal Sharma and Difan Liu and  Subhransu Maji and Evangelos Kalogerakis and Siddhartha Chaudhuri and Radomír Měch. ParSeNet: A Parametric Surface Fitting Network for 3D Point Clouds. European Conference on Computer Vision (ECCV), 2020.*
>
> *[7] Panos Achlioptas and Olga Diamanti and Ioannis Mitliagkas and Leonidas Guibas. Learning Representations and Generative Models for 3D Point Clouds. International Conference on Machine Learning (ICML), 2018.*
>
> *[8] Charles R. Qi and Hao Su and Kaichun Mo and Leonidas J. Guibas. Pointnet: Deep learning on point sets for 3d classification and segmentation. IEEE Conference on Computer Vision and Pattern Recognition (CVPR), 2017.*
>
> *[9] Nilesh Kulkarni and  Abhinav Gupta and  Shubham Tulsiani. Canonical Surface Mapping via Geometric Cycle Consistency. IEEE Conference on Computer Vision and Pattern Recognition (CVPR), 2019.*
>
> *[10] Tinghui Zhou and Philipp Krähenbühl and Mathieu Aubry and Qixing Huang and Alexei A. Efros. Learning Dense Correspondence via 3D-guided Cycle Consistency. IEEE Conference on Computer Vision and Pattern Recognition (CVPR), 2016.*
>
> *[11] Tristan Aumentado-Armstrong and Alex Levinshtein and Stavros Tsogkas and Konstantinos G. Derpanis and Allan D. Jepson. Cycle-Consistent Generative Rendering for 2D-3D Modality Translation.  International Conference on 3D Vision (3DV), 2020.*
>
> *[12] Thibault Groueix and Matthew Fisher and Vladimir G. Kim and Bryan C. Russell and Mathieu Aubry. Unsupervised Cycle-consistent Deformation for Shape Matching. Symposium on Geometry Processing (SGP), 2019.*

---

### Author Response · Authors · 2021-08-10
**General Response**

We appreciate the reviewers for recognizing that our paper addresses an important research problem (Reviewer yaub) with a novel approach (Reviewer kS8b, DYH7) and insightful experiments (Reviewer xuCY).
Here we provide common responses to the reviewers’ feedback in the following four aspects: robustness to unaligned point clouds, the intuition behind handling non-genus-zero shapes, clarification on the theoretical argument in Section 3.2, clarification on the implementation of $\alpha$ in $L_{ACD}$ in Eq. (1).


### **(A) Robustness to unaligned point clouds** [kS8b, xuCY, DYH7]
Reviewer kS8b and DYH7 raise issues regarding the rotation range in our unaligned experiment. Reviewer xuCY asks about the intuition behind why our approach is robust to rotation. Reviewer DYH7 and xuCY ask for clarification on the measurement of confidence score for unaligned inputs.

**More comprehensive degrees of rotation:** We provide empirical justification using multiple different standard deviations, varying from 0.1 to 1.0. We report the averaged error between the predicted and the ground-truth keypoints in Table 1. To better illustrate the correspondence accuracy (as shown in Fig3. in the paper), we also report the accuracy when an error is below a given threshold  (0.05/0.15/0.25) in Table 2. In addition, we conduct experiments under uniformly sampled rotation angles in a certain range of degrees. The accuracies are shown in Table 3 under different thresholds similar to Table 2. The results suggest that when the rotation is under range $\pm30^{\circ}$, our model’s accuracy drop is rather moderate($\sim$15%) compared to the other two baselines($\sim$25% and $\sim$35%).

**Intuition for robustness to unaligned inputs:** As stated in the paper L185~188, our designed objectives can always encourage the input point clouds to be mapped, and *overlapped* on the sphere. Unlike [1, 2] where a common part abstraction can be discovered only when shapes are all aligned, we apply common losses (i.e., reconstruction and chamfer losses) that do not need any alignment assumption. While the primitives (i.e., $U_{A}$) are learned, the rotation information will be encoded in the latent shape code.

**Confidence score via Euclidean distance:** We note that the Euclidean distance is measured between a source, and its “cross-reconstructed” shape (which ideally should be identical), rather than between a source and its target. Thus it will not be affected by miss-alignment between two input point clouds. In other words, our latent shape code encodes rotation info so that the reconstructed point cloud remains with the same pose. We will clarify the intuitions and details in the revised paper.


*Table 1: Averaged Error (the lower the better) comparison between Liu et al. [1] and our model’s variants.*

|  | aligned | $\mathcal{N}(0,0.1)$ | $\mathcal{N}(0,0.2)$ | $\mathcal{N}(0,0.3)$ | $\mathcal{N}(0,0.4)$ | $\mathcal{N}(0,0.5)$ | $\mathcal{N}(0,0.6)$ | $\mathcal{N}(0,0.7)$ | $\mathcal{N}(0,0.8)$ | $\mathcal{N}(0,0.9)$ | $\mathcal{N}(0,1.0)$ |
|---|---:|---:|---:|---:|---:|---:|---:|---:|---:|---:|---:|
| Liu et al. | 0.064 | 0.162 | 0.165 | 0.173 | 0.187 | 0.205 | 0.230 | 0.257 | 0.286 | 0.308 | 0.324 |
| Ours ($w/o L_{cross}$) | 0.070 | 0.074 | 0.094 | 0.122 | 0.154 | 0.186 | 0.219 | 0.253 | 0.282 | 0.304 | 0.321 |
| Ours | 0.054 | 0.061 | 0.064 | 0.072 | 0.084 | 0.106 | 0.141 | 0.184 | 0.227 | 0.262 | 0.290 |

*Table 2: Accuracy (numbers in percentage, the higher the better) comparison between Liu et al. [1] and our model’s variants under different error thresholds (0.05/0.15/0.25).*

|  | aligned | $\mathcal{N}(0,0.1)$ | $\mathcal{N}(0,0.2)$ | $\mathcal{N}(0,0.3)$ | $\mathcal{N}(0,0.4)$ | $\mathcal{N}(0,0.5)$ | $\mathcal{N}(0,0.6)$ | $\mathcal{N}(0,0.7)$ | $\mathcal{N}(0,0.8)$ | $\mathcal{N}(0,0.9)$ | $\mathcal{N}(0,1.0)$ |
|---|---:|---:|---:|---:|---:|---:|---:|---:|---:|---:|---:|
| Liu et al. | 48/92/99 | 19/57/77 | 16/55/77 | 14/51/74 | 12/46/70 | 10/40/64 | 8/34/57 | 6/28/49 | 5/23/43 | 4/20/37 | 4/17/34 |
| Ours ($w/o L_{cross}$)| 50/88/97 | 43/88/97 | 34/81/94 | 26/69/88 | 19/56/79 | 14/45/69 | 10/36/60 | 8/29/51 | 6/24/44 | 5/20/39 | 4/17/35 |
| Ours | 63/92/98 | 53/92/98 | 51/90/98 | 48/87/97 | 43/82/95 | 37/73/89 | 29/61/80 | 21/49/69 | 15/38/58 | 12/31/49 | 9/25/43 |


*Table 3: Accuracy (numbers in percentage, the higher the better) comparison between Liu et al. [1] and our model’s variants under different error thresholds (0.05/0.15/0.25).   Rotations are sampled from Uniform distribution with different bounds.*

|  | aligned | $\pm15^{\circ}$ | $\pm30^{\circ}$ | $\pm45^{\circ}$ | $\pm60^{\circ}$ | $\pm75^{\circ}$ | $\pm90^{\circ}$ |
|---|---:|---:|---:|---:|---:|---:|---:|
| Liu et al. | 48/92/99 | 17/56/77 | 13/51/74 | 10/42/67 | 7/32/56 | 5/24/44 | 4/18/35 |
| Ours ($w/o L_{cross}$) | 50/88/97 | 38/85/96 | 25/67/87 | 15/48/73 | 10/34/58 | 6/24/45 | 4/18/36 |
| Ours | 63/92/98 | 52/92/98 | 48/87/97 | 40/78/93 | 28/60/81 | 16/41/62 | 9/26/45 |


### **(B) Intuition behind handling non-genus-zero shapes** [kS8b, xuCY]
Genus-0 issues exist in mesh representation, where topology cannot be changed during deformation, and shapes are usually parametrized via a simple offset ($\Delta$ x,y,z) to a fixed set of vertices on a template. In contrast, our method: (1) deals with point clouds that do not restrict to constrained topology as meshes; (2) contains MLPs that can map each individual point to an arbitrary position in another space, or in other words, folding a surface arbitrarily so that the resulting shapes can be beyond genus-zero. We will emphasize them in the revised paper.

### **(C) Clarification on the theoretical argument in Section 3.2** [xuCY, DYH7]
In Sec 3.2, we want to show that the order of points (shown as colored points within the sphere) in the canonical primitives will be preserved (as opposed to being shuffled) when mapped back to individual shapes. Thus, overlapped points, i.e., $U_{A}$ and $U_{B}$, will be close to each other in the output space, i.e., $p_{A}$ and $p_{B}$, when being concatenated with the same latent code and mapped to the world coordinate. The proof supports the method of finding correspondences at the inference stage (i.e., see *Inference via CPAE* in L211~216).

The key to proving the order of points is preserved is to prove the decoder (MLPs) is a continuous function that can preserve the local neighborhoods in the output space (i.e., spatially, the output point presents smoothly-transitioned coordinates). The proof includes 1. the linear projection layer is affine, and 2. the ReLU is piecewise linear [3]. An MLP with 1 and 2 satisfies the continuity requirement [4]. In the paper, we focused on 1 due to limited space. We will add the full mathematical proposition and clarify the confusion in the revised paper.


### **(D) Clarification on the implementation of $\alpha$ in $L_{ACD}$** [yaub, DYH7]
Our training procedure is simple. As mentioned in L254~256, we use a two-stage training paradigm where $\alpha$ is directly set to zero in the second stage.  We found the network is not sensitive to the way $\alpha$ is reduced and such a simple policy works well. We will clarify that in our revised version.

---

*[1] Feng Liu and Xiaoming Liu. Learning implicit functions for topology-varying dense 3d shape correspondence.In Advances in Neural Information Processing Systems (NeurIPS), 2020.*

*[2] Zhiqin Chen and  Kangxue Yin and  Matthew Fisher and  Siddhartha Chaudhuri and  Hao Zhang. BAE-NET: Branched Autoencoder for Shape Co-Segmentation. In Proceedings of the IEEE International Conference on Computer Vision (ICCV), 2019*

*[3] Raman Arora and Amitabh Basu and Poorya Mianjy and Anirbit Mukherjee. Understanding Deep Neural Networks with Rectified Linear Units. In International Conference on Learning Representations (ICLR), 2018*

*[4] Thibault Groueix and Matthew Fisherand Vladimir G. Kim and Bryan C. Russell and Mathieu Aubry. A papier-mâché approach to learning 3d surface generation. In Proceedings of the IEEE Conference on Computer Vision and Pattern Recognition (CVPR), 2018*

---

### Decision · Program_Chairs · 2021-09-27

**Decision:**

Accept (Poster)

**Comment:**

This submission received 4 positive final ratings: 6, 6, 7, 6.
On the positive side, reviewers acknowledged importance of the problem, originality of contributions (asymmetric Chamfer distance loss) an overall meaningful and well motivated approach, well executed ablation studies and strong performance. At the same time, they expressed concerns with the method's complexity (multiple loss terms that need balancing), requested additional clarifications (on training details, rotational invariance) and suggested to test the method on a real-world dataset. These concerns were mostly addressed in the rebuttal.
The final recommendation is therefore to accept for poster presentation.